# OpenReview forum: "RewardAnything: Generalizable Principle-Following Reward Models"
_ICLR.cc/2026/Conference — Submitted to ICLR 2026_

### Official Review · Reviewer_rLRA · 2025-10-14

**Soundness:** 3
**Presentation:** 3
**Contribution:** 1
**Rating:** 2
**Confidence:** 4

**Summary:**

The paper introduces REWARDANYTHING, a reward modeling framework that enables language models to follow arbitrary, natural language-specified principles at inference time. It functions as a language model judge trained on preference data using Group Relative Preference Optimization (GRPO). To evaluate this capability, the authors propose RABENCH, a new benchmark designed to assess how well reward models generalize to diverse and previously unseen evaluation principles. Their results show that existing reward models struggle with such generalization, whereas REWARDANYTHING performs strongly, often matching or surpassing the performance of models like GPT-4.1 on both traditional benchmarks and tasks requiring principle generalization.

**Strengths:**

1. The paper addresses an important challenge in reward modeling: enabling flexible alignment of language models through natural language principles rather than static preference datasets.
2. The paper is mostly well-written and organized. Key methods like GRPO and the listwise reward structure are clearly explained. The paper is technically competent, and the proposed method is well-implemented.
3. The proposed dataset and benchmark are open source, which could be useful tools for future research in this space.

**Weaknesses:**

1. The paper lacks in-depth analysis regarding whether the model’s outputs, when conditioned on natural language principles, truly align with those principles. Section 4.2 provides only a very high-level description of how the quality of the preference data is ensured, offering no details on how each component of the dataset, such as the principles and the chosen/rejected responses, is evaluated. There is also no information about the annotator recruitment process or qualification criteria.
2. The paper does not clearly explain how the principles were curated. Since the dataset is a critical component of this work, a more thorough analysis of its quality is necessary. I only found a vague statement on line 163: “We manually curated 200 distinct principles,” along with some generic analysis in Appendix A. However, there is no clear explanation of how the principles were curated. Did you write the principles yourself, or were they sourced from existing datasets?
3. The idea of using LLMs as generative reward models conditioned on natural language and producing structured reasoning has already been explored in prior work [1, 2, 4]. Among these, [1] appears to be the most similar to the proposed approach. Based on my reading, the primary methodological difference is the use of GRPO for training the reward model in this work, where as [1] used pairwise reward model loss. Although [1] is listed in the references, it is neither discussed nor even mentioned in the main text, which is a significant omission. Please correct me if I’m mistaken. Furthermore, similar methods are already widely used in alignment research. For example, [3] aligns models using what they term “specs,” which are effectively equivalent to the natural language principles used here. Likewise, [5] introduces “checklist-guided evaluation” for alignment, where the checklists are derived from in-situ user interactions and feedback. The distinctions claimed in this work, such as GRPO training or explicit principle-following, appear superficial without a direct comparison to these existing approaches.

---
Related Work

[1] Improving Context-Aware Preference Modeling for Language Models, NeurIPS 2024, https://openreview.net/forum?id=52r4XJYzjg

[2] CARMO: Dynamic Criteria Generation for Context-Aware Reward Modelling, ACL 2025, https://arxiv.org/abs/2410.21545

[3] Deliberative Alignment: Reasoning Enables Safer Language Models, OpenAI 2024, https://arxiv.org/abs/2412.16339

[4] Generative Reward Models, Arxiv 2024, https://arxiv.org/abs/2410.12832

[5] WildFeedback: Aligning LLMs With In-situ User Interactions And Feedback, Arxiv 2024, https://arxiv.org/abs/2408.15549

**Questions:**

1. How each component of the dataset, such as the principles and the chosen/rejected responses, is evaluated?
2. How the principles were curated or how they influenced the generation of preference data and the training process?
3. How does your work differ from
- Improving Context-Aware Preference Modeling for Language Models, NeurIPS 2024, https://openreview.net/forum?id=52r4XJYzjg
- Deliberative Alignment: Reasoning Enables Safer Language Models, OpenAI 2024, https://arxiv.org/abs/2412.16339?

---

> ### Author Response · Authors · 2025-11-20
> **Rebuttal for rLRA (Part 1)**
>
> Thank you for the detailed review and for engaging deeply with our work. We appreciate the opportunity to address your concerns:
>
> **W1: Lack of in-depth analysis on data quality, principle alignment, and annotation process.** You raise valid concerns that Section 4.2 provides high-level descriptions without sufficient detail on how dataset components are evaluated and validated.
>
> Here we want to provide much more detailed information. **For RABench construction, we implemented a rigorous two-stage quality assurance process.** First, we used the same synthetic generation pipeline as our training data, employing four state-of-the-art LLM judges (Claude-3.7 Sonnet, GPT-4.1, DeepSeek-V3, Gemini 2.5 Pro) with our consensus algorithm (Algorithm 1 in Appendix) to synthesize initial judgments, but on different prompts and principles, and we explicitly ensured there's no overlap or similar prompts / principles between the trainset and the testset, by running extra comparison and matching. The algorithmic consensus already filters out many inconsistent or low-quality judgments by requiring agreement from at least K=3 out of 4 judges on pairwise preferences.
>
> **Second, and most critically, we conducted comprehensive human verification of the entire RABench.** Two authors independently annotated every single instance in RABench. Each annotator was presented with complete 4-tuples: (principle, prompt, candidate responses from different models, consensus ranking with scores). The annotation task was binary: determine whether the 4-tuple is valid or invalid. **A 4-tuple is marked invalid if any of:** a) The principle is not suitable or applicable to the prompt (e.g., requiring a specific conversational tone for a pure coding problem) b) The consensus ranking does not appropriately reflect adherence to the principle (e.g., when a response clearly following the principle is ranked lower than one that doesn't) c) The responses themselves are problematic to evaluate.
>
> **The inter-annotator agreement reached 89% indicating good reliability**, especially considering the subjective and nuanced nature of this judgment task. **We retained only the 1K instances where both annotators agreed on validity**, ensuring RABench reflects genuine human judgment about principle adherence. This rigorous human validation distinguishes RABench from purely synthetic benchmarks and provides strong evidence that our model genuinely learns principle-following capabilities that align with human understanding.
>
> Regarding annotator qualifications: the two annotators are co-authors of this paper with extensive NLP research experience and deep familiarity with reward modeling, preference learning, and LLM evaluation methodologies. While we acknowledge this represents a limited annotation pool, the high agreement rate and the nature of the task (validating whether rankings reflect stated principles) suggest the judgments are reliable. We will add all these critical details to Section 4.2 in the revision.
>
> It's also worth noting that even the most widely-adopted reward model benchmark, **like RewardBench-v2 and RM-Bench, is built primarily on synthetic data and is considered highly effective by the community**. Our additional human verification step actually places RABench on stronger methodological footing than many existing benchmarks.
>
>
>
> **W2: Insufficient explanation of principle curation process.** Thank you for pointing this out. We will substantially expand this explanation. **Here is our complete principle curation methodology:**
>
> We began by defining a taxonomy of text quality dimensions, resulting in five fundamental categories shown in Table 1: Content (what information to present), Logic (reasoning and coherence), Structure (organization and layout), Style (linguistic choices), and Tone (emotional register and affect). This taxonomy was developed analysis of existing evaluation rubrics.
>
> For each category, we manually wrote diverse seed principles at varying levels of specificity and abstraction. **We explicitly aimed to avoid overly generic principles** (e.g., "be helpful and accurate") that are too broad to meaningfully guide evaluation. Instead, we focused on **specific, concrete, and challenging requirements**. Then we use these seed principles to write more with the help of LLMs, by asking LLMs to generate more candidates and manually write new principles inspired by these synthetic candidates. We used embedding-based similarity detection (faiss) to identify and remove near-duplicate principles. We then manually reviewed all remaining candidates, filtering for clarity, specificity, and practical applicability. **Principles that were too vague, too complex to operationalize were removed.**
>
> We will add this detailed methodology to the paper, likely in a dedicated subsection in the appendix, and provide representative examples of principles at different specificity levels.

---

> ### Author Response · Authors · 2025-11-20
> **Rebuttal for rLRA (Part 2)**
>
> **W3: Comparison with prior work and distinguishing our contributions**
>
> We sincerely appreciate your careful reading and the valuable references you've provided. We acknowledge that we should have provided more thorough discussion of related work, particularly [1] and [3]. We address each of your concerns below and will substantially revise the Related Work section accordingly.
>
> **On model availability and direct comparison.** Although there are works like [1], [2], and SALMON (which is very similar to [1] and is in our Related Work section) that propose to instruct reward models, **none of these release the reward model weights or scripts to replicate their reward models**, so we unfortunately cannot do a direct comparison in our RABench or RewardBench, RM-Bench. We would genuinely welcome incorporating a direct comparison if there's any open-source approach we are missing. However, as these reward models use reward or SFT loss, they can be considered very similar to the SFT settings in our ablation experiment (Table 4), which is significantly worse than our approach with RL (62.3% vs 80.5%). This 18-point gap demonstrates that training methodology matters substantially, and that principle-following is a non-trivial capability worth systematic study.
>
> **[1]**'s research question is: **"How can RMs better evaluate preference given specific context?"** They focus on building context-aware preference evaluation where context resolves inherent ambiguity in underspecified queries. Their canonical example is "Why is the sky blue?" where preference legitimately reverses between "Be scientific" (favoring technical explanation) vs "Explain like I'm five" (favoring simple explanation). The context reveals who the user is or what situation they're in, fundamentally changing what is preferred. Such context is highly problem-specific, and requires quality assurance on each context generated for every problem. They develop methods to train RMs that can evaluate such context-dependent preferences.
>
> Our research questions are: **"How well do existing RMs and general LLMs adapt to specific principles or rubrics? How can we make RMs better adapt to complex principles? How does such adaptation benefit downstream RLHF workflows?"** We focus on measuring and improving the principle-adaptation capability of reward models. Our principles (e.g., "encourage logical reasoning with clear premises," "prioritize safety over detail") are explicit evaluation criteria or rubrics that should apply consistently across tasks. We systematically measure how well current models follow such principles (RABench), develop methods to improve this capability (RL training with different objectives), and demonstrate downstream RLHF benefits (case study in Section 6.3).
>
> Why this distinction matters: Works like SALMON and [1] advances our understanding of context-aware preference evaluation, which is critical for handling diverse user needs. Our work advances our understanding of principle-following as a generalizable capability, which is critical for flexible, interpretable alignment that can adapt to new evaluation criteria without retraining.
>
> Our SFT vs GRPO comparison demonstrates that **making RMs adapt to complex principles is non-trivial**. Simple fine-tuning on principle-conditioned preference data does not automatically yield strong principle-following behavior. This finding is important because it shows that future work on context-aware, rubric or principle-based reward models still has substantial novelty to contribute: the adaptation problem is not solved by straightforward supervised learning.
>
> **[3]** is **domain-specific to safety alignment**. As the title indicates, it uses specifications derived from safety policies for categories. [3]'s research question is: **"How can we use reasoning and explicit safety policies to improve safety alignment?"** They develop methods for safety-specific alignment using specifications derived from safety policies (e.g.: erotic, self-harm, violence, extremism). Their specifications define what content is "allowed," "disallowed," or "requires safe completion" for safety scenarios. [3] makes critical contributions to safety alignment through deliberative reasoning. Our work contributes a general framework for principle-following reward models with systematic evaluation of generalization capabilities, which extends beyond safety to any domain where explicit evaluation criteria are valuable.

---

> ### Author Response · Authors · 2025-11-20
> **Rebuttal for rLRA (Part 3)**
>
> **[2]** We thank you for this reference. Based on the paper, CARMO prompts LLMs to generate dynamic evaluation criteria to mitigate reward hacking, and could be considered similar to "Inference-Time Scaling for Generalist Reward Modeling" which we discussed in our Related Work. Such methods face the challenge that RMs must "guess" the underlying criteria to make judgments, where the generated criteria might introduce new issues or biases. In contrast, our approach uses explicit, predefined principles with systematic evaluation of whether models can follow them accurately. We will add appropriate discussion of CARMO in our revision, and would greatly appreciate it if you could clarify any specific aspects of CARMO that you believe we have missed, as this will help us provide the most relevant discussion.
>
> **[4] and [5]** [4] demonstrates the success of generative reward models and LLM-as-a-judge approaches, which aligns with our choice of generative architecture. [5] (WildFeedback) introduces a framework to automatically create preference datasets with checklists derived from in-situ user interactions. While [5] focuses on automatically generating evaluation criteria from user feedback, our work focuses on measuring and improving models' ability to follow explicitly specified principles. These works represent complementary approaches to making evaluation more flexible and aligned with diverse preferences. We sincerely thank you for pushing us to clarify these distinctions. We believe our work, [1], and [3] represent complementary and important directions for making reward models more flexible, interpretable, and aligned with diverse human preferences.
>
>
>
> We sincerely thank you again for your thorough and constructive review, which has significantly strengthened our paper. We will incorporate all the detailed clarifications provided above into our revision, including the expanded discussion of data quality assurance in Section 4.2, the comprehensive principle curation methodology (likely in a dedicated appendix subsection), and substantially revised Related Work section with careful positioning relative to [1], [2], [3], [4], and [5]. We believe these additions directly address your core concerns about dataset quality validation, principle generation process, and distinguishing our contributions from prior work. We hope this detailed response clarifies our methodology and contributions, and we would be grateful if you could reconsider your assessment in light of these clarifications. If there are any remaining concerns or aspects you feel require further justification, we are very happy to engage in continued discussion.

---

> ### Comment · Reviewer_rLRA · 2025-11-20
>
> Thank you for the detailed rebuttal and for taking the time to expand on several aspects of the work. I still have several concerns.
>
> ---
>
> **Concerns about RABench, deduplication, and generalization**
>
> RABench is presented as a held-out evaluation set, but both the training data and the benchmark are generated through essentially the same synthetic pipeline. Even with the separation of principle IDs and prompt sources, the overall process is very similar, which makes the evaluation feel more in-domain than truly out-of-distribution. It would help to include concrete examples of principles or prompts that you consider genuinely outside the training distribution.
>
> The paper states that contamination was avoided, but it does not describe how near-duplicate or semantically overlapping principles were detected or filtered. The rebuttal mentions extra similarity checks, yet these procedures do not appear in the submitted draft. Please clarify how principle overlap was measured and how you ensured that the full set of 200 principles represents distinct evaluation criteria.
>
> I also cannot inspect the principle list, filtering rules, or generation scripts because the anonymous GitHub link in the submission has expired. If alternative materials exist, please provide them so that the construction of RABench can be verified.
>
> Finally, stronger evidence of generalization to truly unseen principles would improve the paper. All evaluation principles come from the same manually curated pool, which makes it unclear how the model performs on principles written independently or in different styles or abstraction levels. Examples or experiments along these lines would help support the central claim.
>
> ---
>
> **Concerns about novelty and relation to prior work**
>
> Your expanded discussion of related work is appreciated, but the novelty still feels limited. The core design, which is a principle-conditioned generative reward model trained with listwise data and an RL objective, is very similar to earlier instructable and criteria-conditioned reward modeling approaches. At present, the main difference seems to be the construction of a new dataset rather than a clearly new modeling idea.
>
> If there is a deeper reason why your framework should generalize better than earlier instructable RMs, it is not yet articulated. One concrete type of evidence you could provide is that the principles you use are more diverse or higher quality than those in previous work, which would directly support better generalization. The current draft does not present this.
>
> I also note that the paper *Generative Reward Models* [4] was rejected at ICLR 2025 last year in part for a similar concern, where reviewers expressed significant concerns regarding novelty. Clear evidence of what is fundamentally new here, beyond the specific dataset, would significantly strengthen the contribution.
>
> ---
> I am happy to hear more of your thoughts or any additional information that can address these concerns.

---

> > ### Author Response · Authors · 2025-11-25
> > **Response to Reviewer rLRA (Follow-up, Part 3)**
> >
> > **Summary of our follow-up**
> >
> > To directly address your two core concerns: On novelty, the contribution is not *just* the dataset but Group Relative Preference Learning, a fundamentally different training paradigm that teaches reward models to discriminate relative quality given specified principles through RL exploration rather than memorizing preferences through SFT. The 18-point gap between SFT and GRPO on identical data is direct empirical proof, and no prior work in context-aware or instructable reward modeling has demonstrated this generalization capability or performance gain from training methodology alone (excl. data strategies). On generalization, Section 6.3 provides exactly the out-of-distribution experiment you request, with a manually crafted, complex principle outside our training distribution, validated on downstream alignment tasks with real world safety implications. RewardBench-v2 results further confirm generalization to an independently constructed external benchmark (guaranteed by release time) where we perform comparably to frontier models like Claude Sonnet 3.7 and GPT-4.1.
> >
> > We sincerely hope these clarifications address your remaining concerns. We have invested significant effort in providing concrete, verifiable evidence for each claim. If there are specific aspects you would like us to elaborate further, we remain fully engaged and grateful for the opportunity to strengthen this work through your rigorous review.

---

> ### Author Response · Authors · 2025-11-25
> **Response to Reviewer rLRA (Follow-up, Part 1)**
>
> Thank you for your engagement and for giving us the opportunity to clarify what we believe are fundamental misunderstandings about the nature of our contribution. We address each concern with concrete evidence below.
>
> ## **Regarding the expired anonymized codebase**
>
> Thank you for pointing this out, this is due to an automatically configurated expiration by Anonymous GitHub. We have now resubmit our codebase to this new link for your reference: https://anonymous.4open.science/r/iclr-submission-915/
>
> ## **Novelty Clarifications**
>
> **On Methodological Novelty: Group Relative Preference Learning is Fundamentally Different**
>
> We respectfully but firmly disagree that our work merely *"applies GRPO to reward models"* or that *"the main difference is the construction of a new dataset."* This interpretation misses the core methodological insight of our work. Allow us to explain precisely why **Group Relative Preference Learning represents a fundamentally different training paradigm**.
>
> The landscape of reward model training today consists of two mainstream approaches. The first is the *Bradley-Terry* paradigm, where discriminative reward models are trained as sequence classifiers using a regression-like objective that maximizes scoring differences between chosen and rejected responses. The second is the *SFT* paradigm, where generative models (including those in context-aware RM works you cite) are fine-tuned to memorize pairwise or pointwise judgments, sometimes with reasoning chains distilled from stronger judges, sometimes with contexts distilled from stronger models, these all require the RM to remember precisely whats in their training data.
>
> **Our approach is neither.** In Group Relative Preference Learning, **we do not use GRPO to elicit reasoning capabilities (as the hundreds of recent GRPO papers do for math / code / etc. reasoning)**. Instead, we use reinforcement learning to teach the reward model a fundamentally different objective: **learning to discriminate relative ordering and absolute quality differences among groups of responses through its own exploratory rollouts, given a clear principle**. This is a paradigm shift from "memorizing preferences" to "learning to judge and generalize to diverse principles and preferences."
>
> Consider Section 5.2 of our paper, where we define the *accuracy* reward (Equation 4). The *format* reward component ensures the model produces non-contradictory judgments, addressing a critical failure mode where backbone models without RL training frequently assign higher scores to response A while simultaneously ranking B as better. The accuracy reward component weights each misordering by the magnitude of the quality gap: incorrectly ordering two responses with a 4-point score difference is penalized far more heavily than swapping two responses with similar scores. **Both terms are dense reward signals that effectively teaches the model to understand relative quality magnitudes, not just binary preferences.** If we were to remove both reward components, RL training would become meaningless as there would be no learning signal, which is also why our ablations in Table 4 show modest impact when removing only one component at a time: the remaining term still provides modest supervision.
>
> **The empirical evidence is unambiguous.** Table 4 in our paper shows that when we train on the exact same data using SFT (which precisely represents the training procedure of context-aware RM works like [1]), performance drops from 80.5% to 62.3% on RABench. This 18-point gap demonstrates conclusively that the contribution is not the data but the training methodology. If context-aware approaches using SFT were sufficient, this gap would not exist. Furthermore, converting our listwise data to pairwise comparisons (mimicking the data format of [1], and also a lot of existing literature) degrades performance from 80.5% to 73.2%, demonstrating that learning relative preferences as a group provides additional signal beyond pairwise decomposition.
>
> This is analogous to how reinforcement learning has reshaped math/code reasoning capabilities in LLMs: **we are using RL to enable generalization to diverse evaluation criteria, rather than having the model memorize a large catalog of context-principle-judgment tuples during training (e.g. the context-aware RM you mentioned). To the best of our knowledge, this is something that no previous or concurrent work have done.** **The model learns how to judge according to principles, not what judgments to produce for specific principles. This distinction is fundamental and non-trivial that should not be automatically ignored.**

---

> ### Author Response · Authors · 2025-11-25
> **Response to Reviewer rLRA (Follow-up, Part 2)**
>
> **On Generalization Evidence: Strong Results on Truly Out-of-Distribution Settings**
>
> You raise a valid concern about whether RABench evaluation is truly out-of-distribution. We offer the following concrete pieces of evidence that address this directly.
>
> **First, our case study in Section 6.3 provides exactly the experiment you request.** The principle used in this case study was manually crafted specifically to be outside our training distribution. It combines multiple conflicting objectives (safety, warmth, humor, helpfulness) with explicit prioritization, and it was written in a different style than our 200 curated principles. Critically, we used this novel principle to align an LLM (Qwen3-8B) via GRPO with RewardAnything as the sole reward source, then evaluated on established benchmarks (XSTest for safety, MT-Bench for quality). The significant improvements over both the base model and a Skywork-RM aligned baseline demonstrate that our model genuinely generalizes to novel, human-specified principles in real downstream tasks, not just on benchmarks.
>
> **Second**, our newly reported **RewardBench-v2** results (provided to Reviewer zuWy) demonstrate generalization to an entirely external benchmark. **Every single model in this table is trained before the release of RewardBench-V2**, so it's clearly a OOD scenario in terms of distribution or prompt and preference, and we achieve 74.5% overall accuracy, placing us between Claude Sonnet 3.7 (75.4%) and GPT-4.1 (72.3%). Notably, we achieve 47.5% on the "Precise Instruction Following" split compared to GPT-4.1's 39.7%, directly validating our principle-following capability on an independently constructed benchmark with principles we never trained on.
>
> | **Model**                        | **Type**       | **Overall** | **Factuality** | **Precise IF** | **Math** | **Safety** | **Focus** | **Ties** |
> | -------------------------------- | -------------- | ----------- | -------------- | -------------- | -------- | ---------- | --------- | -------- |
> | Claude Sonnet 3.7 (20250219)     | Generative     | 75.4        | 73.3           | 54.4           | 75.0     | 90.3       | 92.1      | 67.2     |
> | **RewardAnything-8B**            | Generative     | 74.5        | 66.7           | 47.5           | 80.3     | 84.9       | 84.2      | 83.3     |
> | GPT-4.1 (20250414)               | Generative     | 72.3        | 82.9           | 39.7           | 65.2     | 87.3       | 73.4      | 85.4     |
> | Skywork-Reward-Llama-3.1-8B-v0.2 | Discriminative | 71.8        | 69.7           | 40.6           | 60.1     | 94.2       | 94.1      | 71.7     |
> | GPT-4o (20240806)                | Generative     | 64.9        | 56.8           | 33.1           | 62.3     | 86.2       | 72.9      | 78.2     |
> | GPT-4o-mini (20240718)           | Generative     | 58.0        | 41.1           | 34.4           | 51.9     | 76.7       | 74.1      | 69.6     |
> | GRM-llama3-8B-sftreg             | Discriminative | 60.9        | 61.9           | 38.8           | 57.9     | 78.7       | 68.3      | 59.8     |
> | ArmoRM-Llama3-8B-v0.1            | Discriminative | 66.5        | 65.7           | 41.9           | 66.1     | 82.2       | 76.6      | 66.3     |
>
>
>
> **On Comparison with Prior Work**
>
> You specifically ask how our work differs from [1] (Improving Context-Aware Preference Modeling) and [3] (Deliberative Alignment). The distinction is not superficial, however we totally understand if you hold your opinion on the high-level novelty claims. **However, most importantly, none of these prior works release model weights or replication scripts, making direct comparison impossible, and we have not yet found any direct reproduced result or attempt to these works.** This should not be automatically ignored, especially at a venue like ICLR where open science is encouraged.
>
> However, since their training procedures are SFT-based, they can be considered analogous to our "GRPO to SFT" ablation, which underperforms our full method by 18 points. This gap is the empirical signature of our methodological contribution. Note that in the domain of reward modeling, where the vast majority of papers focus on curating preference data rather than innovating training techniques (including widely-used works like Skywork-Reward and Skywork-Reward-V2, and everything on RewardBench's official leaderboard), a methodological contribution that produces an 18-point improvement represents significant original contribution.

---

> ### Comment · Reviewer_rLRA · 2025-11-25
>
> Thanks for the detailed response. I’ve read through all of it carefully and have improved my score. The added reward bench v2 experiment is helpful, but I’m still not sure it addresses my concern about generalizing to unseen principles. What rubrics or principles did you provide to your GenRM for reward bench v2, or is it operating purely zero-shot? If it is zero-shot, then how can the model be said to generalize to unseen principles? In that case, isn’t it just pure reward modeling without any conditioning on principles? Please correct me if I’m mistaken.

---

> > ### Author Response · Authors · 2025-11-26
> > **Response to Reviewer rLRA (Follow-up II, Part 2)**
> >
> > | Principle Content                                            | Precise IF Acc |
> > | ------------------------------------------------------------ | -------------- |
> > | **IFFocus**: If the user has specified required formatting, logical structures or request specific patterns in the response, make sure to focus on that requirement as it's the most important. Precise instruction-following is very important. | **51.25%**     |
> > | **Original+IFFocus**                                         | **48.75%**     |
> > | **Original**: You should choose the assistant that follows the user's instructions and answers the user's question best. Your evaluation should consider factors such as the helpfulness, relevance, accuracy, depth, creativity, and level of detail of their responses. | **47.50%**     |
> > | **Original+IFignored**: {Original} + "If the user has specified required formatting, logical structures or request specific patterns in the response, make sure to ignore these and use only the above principles." | **45.00%**     |
> > | **Original (concise)**: You should choose the assistant that follows the user's instructions and answers the user's question best. | **44.37%**     |
> > | *(Empty string)*                                             | **41.77%**     |
> >
> > These results reveal several important findings: a) the fact that IFFocus outperforms Original+IFFocus indicates that the general goals in the original prompt can interfere with attending to user-specific instructions when both are present. b) Original+IFFocus outperforming Original demonstrates that explicitly directing attention to user instructions improves performance, confirming that user-embedded criteria are important evaluation factors. c) using vague or no principle substantially degrades performance, showing that RewardAnything benefits from explicit principle guidance even when the actual evaluation criteria are embedded in the problem text. Most tellingly, when we explicitly instruct the model to ignore user formatting requirements (Original+IFignored), performance drops below the Original baseline, confirming that RewardAnything does attend to and utilize the implicit principles within problem texts.
> >
> > These findings collectively demonstrate that the Precise IF requirements cannot be captured by general preference criteria alone and should indeed be considered novel evaluation principles. Our model's ability to improve when directed to focus on these in-context requirements validates that it has learned generalizable principle-following behavior.
> >
> > ## On Failure Mode Analysis
> >
> > We also recognize that you might consider these scores are not great enough, even though frontier closed source LLMs achieve ~54% acc (though we note that random guessing would achieve approximately 25%). Therefore, we are now also manually studying the failure modes of our reward model on this exact split to understand the relatively modest absolute scores. Upon manual inspection of failed instances, we identified the following primary failure categories:
> >
> > | Failure Type                  | %    | Example                                                      |
> > | ---- | ---- | ---- |
> > | Exact counting | ~15% | "Write a .... The response must contain between 158 and 171 words." |
> > | Punc./indent/special char. requirements      | ~12% | "... Please use an emoji at the end of every sentence." |
> > | Nested structure depth        | ~8%  | "How might an anti-hero ...' Include quotes within quotes within quotes, at least 3 levels deep, alternating between double quotes and single quotes." |
> > | Rare lexical constraints      | ~5%  | "Best languages ...? Include at least 10 single-word palindromes, each at least 5 characters long." |
> >
> > These failure modes represent well-known limitations of LLMs [1,2]. Precise counting of words, characters, or specific indentation patterns is fundamentally constrained by tokenization, which does not preserve character-level or word-level boundaries. Similarly, verifying complex nested structures or finding multiple rare lexical items requires capabilities that current LLM backbones struggle with. These are limitations of the backbone model's architecture rather than failures of principle-following capability.
> >
> > We are deeply grateful for your insightful questions throughout this discussion. We believe these clarifications, along with the new ablation experiments and failure mode analysis, comprehensively address your concerns about generalization to unseen principles. Given these additional results and explanations, we respectfully ask whether you might consider further improving your score. We remain fully committed to addressing any remaining concerns you may have, and we are happy to provide further clarifications if needed. Thank you again for your dedication.
> >
> > [1] Number Cookbook: Number Understanding of Language Models and How to Improve It ICLR'25
> > [2] Why Do Large Language Models (LLMs) Struggle to Count Letters? arXiv 2412.18626

---

> ### Author Response · Authors · 2025-11-26
> **Response to Reviewer rLRA (Follow-up II, Part 1)**
>
> Thank you sincerely for your continued engagement with our work and for improving your score. We truly appreciate the time and effort you have invested in understanding our contributions, and your questions have helped us strengthen the paper significantly. Your question about RewardBench-v2 prompting is excellent and deserves a thorough answer. We would like to address this from two perspectives: **1), how RewardBench-v2 differs from v1 in what it evaluates, and 2) how it specifically tests principle-following capabilities.**
>
> ## 1) On RewardBench-v2's more aligned evaluation goals
>
> Compared with RewardBench v1, RewardBench-v2 was designed specifically to evaluate "challenging, accuracy-based benchmarks correlated with downstream post-training evaluation," this is actually a similar goal to v1, but more aligned, or closer to the goal, rather than format and presentation style related shortcuts or spurious correlations. The Math, Safety, and Focus splits are improved versions over v1, while Factuality, Precise IF, and Ties introduce new challenges: factual accuracy assessment, precise instruction following, and identifying ties among similar-quality responses respectively. This redesign matters because **many previous reward models were overfit to v1's spurious correlations, which is confirmed by the v2 benchmark results themselves in the RewardBenchV2 paper**. This is also noted in the RM-Bench paper and validated in our own experiments (Table 3 4 and Appendix A), traditional RMs often exploit superficial cues like response length or formatting rather than actual quality. RewardBench-v2 was specifically constructed to eliminate these shortcuts, making accuracy the primary factor rather than presentation style. In this sense, the "real, intended accuracy" that v2 measures represents a somewhat new evaluation scenario that previous RMs were not genuinely optimized for, even though it was the original intent of v1.
>
> ## 2) On the Precise IF Split as Principle-Following Evaluation
>
> More importantly for your question, the Precise IF split directly tests the ability of reward models to judge whether responses follow precise, explicit instructions. These instructions function similarly as principles do in our framework, as they specify evaluation criteria that the model must recognize and apply. The key difference is that these criteria appear within the problem text itself, either explicitly stated or implicitly required. For example:
>
> > "Political ideologies of the Socialist Revolutionary Workers' Party. Ensure that the ratio of sentence types (declarative, interrogative, exclamatory) is balanced."
> >
> > "Write a fictional timeline of an alternate universe where the Americas were only discovered in the 1900s. Create stairs by incrementally indenting each new line in your response."
> >
> > "If humans made of ricin? The response should include at least 21 pronouns."
>
> These requirements are human-crafted evaluation criteria that RewardAnything has never seen. They represent truly out-of-distribution principles that test whether our model has learned generalizable criterion-following capabilities.
>
> ## On How We Prompt RewardAnything
>
> Regarding our prompting strategy, we adapted the prompt from RewardBench's official evaluation script (rewardbench/generative_v2.py), removing instructions specific to that script's formatting requirements. The core principle we use is: "You should choose the assistant that follows the user's instructions and answers the user's question best. Your evaluation should consider factors such as the **helpfulness, relevance, accuracy, depth, creativity, and level of detail** of their responses."
>
> However, we recognized that you might question whether this general principle could conflict with the specific requirements embedded in the problem text, and whether RewardAnything appropriately prioritizes explicit user instructions over our provided principle. To address this directly, we ran additional experiments examining how different principle formulations affect performance on the Precise IF split, as there might exist multiple goals:

---

> ### Comment · Reviewer_rLRA · 2025-11-26
>
> Thanks for your detailed response, and sorry for asking so many questions. Could you show the Qwen 3 8B results for Reward Bench V2, similar to how you did for the other experiments? Since Qwen 3 8B is the backbone model of RewardAnything 8B, I’d like to see how much improvement was achieved. I haven’t been able to find any public results on Qwen 3 8B on this benchmark. The closest is Skywork-Reward-V2-Qwen3-8B [1], which reports an average performance of 78.4% on Reward Bench V2 and 50% on IF (about 3% higher than RewardAnything 8B). If you can show that RewardAnything 8B meaningfully improves over its backbone Qwen 3 8B, I’ll raise my score.
>
> [1] Skywork-Reward-V2: Scaling Preference Data Curation via Human-AI Synergy

---

> > ### Author Response · Authors · 2025-11-27
> > **Response to Reviewer rLRA (Follow-up III)**
> >
> > Thank you for your continued engagement and for these excellent questions. Please do not apologize for asking questions: your thorough review has significantly improved the quality of our work, and we are genuinely grateful for the time you have invested. We are happy to provide the Qwen3-8B backbone comparison you requested.
> >
> > We ran the same experiments on Qwen3-8B (our backbone model) to directly measure the improvement achieved by RewardAnything-8B. Notably, our table includes only models trained before RBv2, so there is no possibility of data contamination for any of the results reported.
> >
> > | **Model** | **Type** | **Overall** | **Factuality** | **Precise IF** | **Math** | **Safety** | **Focus** | **Ties** |
> > | -------------------------------- | -------------- | ----------- | -------------- | -------------- | -------- | ---------- | --------- | -------- |
> > | Claude Sonnet 3.7 (20250219) | Generative | 75.4 | 73.3 | 54.4 | 75.0 | 90.3 | 92.1 | 67.2 |
> > | **RewardAnything-8B** | Generative | 74.5 | 66.7 | 47.5 | 80.3 | 84.9 | 84.2 | 83.3 |
> > | GPT-4.1 (20250414) | Generative | 72.3 | 82.9 | 39.7 | 65.2 | 87.3 | 73.4 | 85.4 |
> > | Skywork-Reward-Llama-3.1-8B-v0.2 | Discriminative | 71.8 | 69.7 | 40.6 | 60.1 | 94.2 | 94.1 | 71.7 |
> > | ArmoRM-Llama3-8B-v0.1 | Discriminative | 66.5 | 65.7 | 41.9 | 66.1 | 82.2 | 76.6 | 66.3 |
> > | GPT-4o (20240806) | Generative | 64.9 | 56.8 | 33.1 | 62.3 | 86.2 | 72.9 | 78.2 |
> > | **Qwen3-8B** | Generative | 62.9 | 53.9 | 36.9 | 66.1 | 75.3 | 65.7 | 71.6 |
> > | GRM-llama3-8B-sftreg | Discriminative | 60.9 | 61.9 | 38.8 | 57.9 | 78.7 | 68.3 | 59.8 |
> > | GPT-4o-mini (20240718) | Generative | 58.0 | 41.1 | 34.4 | 51.9 | 76.7 | 74.1 | 69.6 |
> >
> >
> > **Precise IF Ablation:**
> >
> > | Principle Content | RewardAnything-8B | Qwen3-8B |
> > | --------- | ------- | -------- |
> > | IFFocus | **51.25** | 46.87 |
> > | Original+IFFocus | 48.75 | 43.75 |
> > | Original | 47.50 | 36.87 |
> > | Original+IFignored | 45.00 | 38.75 |
> >
> >
> > The results demonstrate substantial and consistent improvements from our training methodology: RewardAnything achieves 74.5 overall acc. compared to Qwen's 62.9. This places RewardAnything-8B between Claude Sonnet 3.7 and GPT-4.1, despite being an 8B parameter open-source model. On the Precise IF split tahts most relevant to our claims, RewardAnything-8B achieves 47.5 compared to Qwen3-8B's 36.9, a 10.6 point improvement.
> >
> > Regarding comparison with Skywork-Reward-V2: You mentioned Skywork-Reward-V2-Qwen3-8B reports 78.4% overall and 50% on Precise IF. We note that this model was released after RewardBench-v2, so direct comparison requires caution regarding potential contamination. Additionally, they disclosed that 26 *million* carefully curated preference pairs by human and LLMs were used for training, while we used only 4 *thousand* fully synthetic listwise preference data (equivalent to 173K pairs). This makes our methodological contribution even more clearly demonstrated: we achieve competitive performance with orders of magnitude less training data through our Group Relative Preference Learning approach.
> >
> >
> > Thank you again for pushing us to provide this important comparison. We believe this evidence strongly supports our contribution, and we respectfully ask if you would consider raising your score as mentioned. We remain available for any further questions or experiments you may find helpful.

---

> ### Comment · Reviewer_rLRA · 2025-11-27
>
> Thanks for the detailed response. I have increased my score. Good luck.

---

### Official Review · Reviewer_GTfZ · 2025-10-26

**Soundness:** 2
**Presentation:** 2
**Contribution:** 2
**Rating:** 4
**Confidence:** 5

**Summary:**

This paper addresses a limitation in traditional Reward Modeling for RLHF: the rigidity of RMs trained on static preference datasets. Current RMs learn implicit preferences and struggle to adapt to new criteria without costly retraining. The authors propose a paradigm shift towards "principle-following reward models," which can dynamically adjust their evaluation criteria based on natural language instructions

**Strengths:**

Timely and Important Problem: The paper tackles a highly relevant problem. The inflexibility and bias of current RMs are major bottlenecks in deploying aligned AI systems.

Efficiency and Interpretability: The listwise ranking approach of REWARDANYTHING is computationally efficient compared to pairwise comparisons, requiring only a single inference call to rank multiple candidates. Furthermore, the model's generation of explicit reasoning steps enhances the interpretability of the reward signal.

**Weaknesses:**

Unfair Comparison Methodology on RM-Bench: The claimed SOTA performance on RM-Bench (Section 6.1, Table 2) appears to rely on a flawed comparison methodology. The authors state they provided REWARDANYTHING with a clear principle prioritizing accuracy, and that "this principle is also passed to other RMs as system prompt" (L355-356). However, the majority of the baselines, particularly the Discriminative Reward Models (e.g., Skywork-Reward, FsfairX, Nemotron), are fine-tuned sequence classifiers or regression models; they are not architecturally designed to interpret system prompts or dynamically adjust their learned preferences based on inference-time instructions. By giving REWARDANYTHING explicit instructions on the evaluation criteria, while the baselines are forced to rely on their implicit learned preferences, the comparison is heavily biased. This undermines the SOTA claim.

Complexity and Stability of GRPO Training: The proposed training method (GRPL) uses GRPO, an online RL algorithm, with a complex, multi-component reward function (Equations 2-4) involving numerous hyperparameters. This is likely significantly more complex and potentially less stable than standard SFT or offline methods like DPO. While the ablation (Table 4) shows GRPO outperforms SFT, the paper lacks analysis of the training costs, stability, and hyperparameter sensitivity of GRPO in this context. The necessity of this complexity over potentially adapting DPO for principle-following is not established.

The Burden of "Principle Engineering": The analysis in Section 6.1 (Figure 3) highlights that performance drops significantly when principles are vague or when multiple objectives lack clear prioritization. This suggests that the success of the method heavily depends on the user's ability to craft high-quality, structured principles. This potentially shifts the challenge from "data collection" to "principle engineering," which presents its own difficulties, and may limit the claimed generalization if the model fails on ambiguous real-world principles.

**Questions:**

In addition to the weakness section, I have the following questions:

How exactly were the discriminative RMs prompted with the principle in the RM-Bench evaluation? Given their architecture, what evidence do you have that they are capable of utilizing this input? How would REWARDANYTHING perform if it were not provided with an explicit principle tailored to the benchmark?

Q2. Regarding W2: Given that the training data relies solely on LLM consensus without human oversight, how do you ensure that REWARDANYTHING is learning generalizable principle-following capabilities rather than overfitting to the specific patterns and biases of the LLM judges (e.g., GPT-4.1) used for data generation?

---

> ### Author Response · Authors · 2025-11-20
> **Rebuttal for GTfZ (Part 1)**
>
> Thank you for the detailed and thoughtful review. We appreciate your recognition that we tackle a timely and important problem with a computationally efficient approach that enhances interpretability. We'd like to address your concerns point by point.
>
> **W1 / Q1: Unfair Comparison on RM-Bench, regarding prompting strategies.** Thank you for the concerns that providing all RMs (including RewardAnything, Gemini, GPT) with explicit principles while discriminative reward models cannot utilize them creates an unfair comparison that undermines our SOTA claim.
>
> We respectfully but firmly disagree with the premise that this comparison is unfair. In fact, we argue that our experimental setup is precisely fair because **all models receive identical prompting: Principle in system prompt, question and evaluated response in user prompt**. The principle is provided in exactly the same way to every model evaluated. **What differs is each model's inherent capability to understand and utilize that information.** Generative models, by their nature, have better natural language understanding and can interpret instructions dynamically. This is not an unfair advantage we artificially created; it is a fundamental capability difference that represents a crucial limitation of existing discriminative reward models.
>
> This limitation is exactly what our work aims to highlight and address. **Discriminative reward models are sequence classifiers trained on regression objectives. They fundamentally cannot interpret natural language instructions or adapt their evaluation criteria at inference time.** Changing the experimental setup to give discriminative models specialized prompting strategies or removing principles from them entirely would actually make the comparison unfair by not evaluating all models under the same real-world deployment conditions.
>
> That said, we recognize this is a key concern for your review. **To specifically address whether the location of principles (system prompt vs. user prompt) significantly impacts discriminative reward models, we conducted additional ablation experiments.** The results strongly support our position:
>
> | Model | Chat | Math | Code | Safety | Easy | Normal | Hard | Overall |
> | -------------------------------- | ------- | ---------- | ---------- | ---- | ------ | ----- | ---- | ---- |
> | Skywork-Reward (Principle in System Prompt) | 69.3 | 62.1 | 53.4 | 96.0 | 89.3 |  75.8 | 52.6 | 72.6 |
> | Skywork-Reward (Principle in User Prompt) |73.3|63.0|52.5|94.3|88.4|75.8|54.3|72.8|
> | FsfairX-LLaMA3-RM (Principle in System Prompt) |67.3|62.8|55.7|91.8|87.4|74.8|52.8|71.7|
> | FsfairX-LLaMA3-RM (Principle in User Prompt) |68.1|62.2|56.2|91.3|87.4|74.8|52.0|71.4|
> | RewardAnything-8B |76.5|89.4|73.7|89.6|88.0|84.8|83.8|85.5|
>
> The evidence is clear: whether principles are placed in system prompts or user prompts makes slight difference to discriminative reward models. This confirms that discriminative models simply cannot leverage principle information regardless of how it's presented, and our experimental setting is not leaning towards ourselves.
>
> It's worth noting that this experimental setting (Principle in User Prompt) closely mirrors RewardBench-v2's "Precise Instruction Following" split, which specifically evaluates whether reward models can adhere to criteria specified in prompts, thus also a valid setting and does not create further problems.
>
> The core insight of our work is that traditional discriminative reward models are locked into implicit preferences learned during training. They cannot adapt to new evaluation criteria without costly retraining. This is not a minor limitation but a fundamental constraint that limits their practical utility. Modern leading LLMs like GPT-4.1 and Gemini do respond meaningfully to principles and system prompts, which is precisely why they're widely used as evaluators despite being general-purpose models. Our work brings this capability to a specialized, efficient reward model suitable for practical RLHF deployment.

---

> ### Author Response · Authors · 2025-11-20
> **Rebuttal for GTfZ (Part 2)**
>
> **W2: Complexity and Stability of GRPO Training.** Thank you for concerns about the complexity of GRPO training with multiple hyperparameters compared to simpler methods like SFT or DPO, questioning whether this added complexity is justified.
>
> We want to clarify several important points. First, reward models typically operate on much smaller datasets than general language model training, where stability and efficiency are paramount concerns. For instance, the widely-used Skywork-Reward-Preference-80K-v0.2 dataset contains only 80K pairs and can be trained within days. Our RewardAnything was trained on 4K listwise preferences (equivalent to 173K pairwise comparisons), completing training in just 23 hours on a single 8×A100-80G machine. This is entirely practical and manageable.
>
> More importantly, the critical consideration for reward models is not training cost but inference cost. Reward models provide feedback signals during RL training where they are called repeatedly for every candidate response. As detailed in Appendix B, our listwise approach requires only a single LLM call to rank all responses, compared to quadratic O(n²) calls for pairwise methods. Recent concurrent works using pairwise comparison with reasoning (such as RM-R1) require 45 inference calls just to rank 10 responses, with each call generating long chain-of-thought sequences. This makes such approaches computationally infeasible for practical RLHF deployment at scale, whereas our method maintains computational efficiency while adding principle-following capability.
>
> Regarding DPO, this requires pairwise comparison preference data, which makes training significantly less efficient compared with the listwise approach we use. Moreover, DPO is an off-policy optimization method widely considered inferior to on-policy counterparts for modern LLMs. For reasoning reward models, this would also require an extra step of distilling other judge models' reasoning chains, bringing significantly more cost to ensure the quality of distilled data. Our ablation in Table 4 provides empirical validation: GRPO substantially outperforms SFT, demonstrating that the added complexity translates to meaningful performance gains in the capability we care about most.
>
> **W3: The Burden of "Principle Engineering."** You suggest that our method shifts the challenge from data collection to principle engineering, potentially limiting claimed generalization when principles are vague or ambiguous.
>
> We'd like to completely reframe this concern. Principle engineering is not a burden; it is an empowering feature that gives users direct control over alignment objectives. Current reward models lock users into a far more burdensome workflow: curating preference datasets with thousands of examples, ensuring annotation quality and consistency, recruiting and training annotators, and completely retraining models whenever objectives need to change (as illustrated in Figure 1). Our approach replaces this entire costly pipeline with natural language specification, which is dramatically more flexible and accessible.
>
> As surveyed in [1], misspecified human preferences, ambiguous choice rules, and unclear preference criteria are the primary sources of reward model failures in practice. RewardAnything directly addresses these fundamental issues by giving users explicit control over preference specification through natural language rather than forcing them to rely on implicit learning from aggregate preferences.
>
> When users provide vague, underspecified principles, RewardAnything's performance naturally degrades to levels comparable to traditional reward models (as shown in Figure 3, also in Appendix A). This is exactly the expected and appropriate behavior. It does not indicate inferiority; rather, it demonstrates that explicit specification enables substantially better performance when done well, with graceful degradation when specification is unclear. Traditional reward models are stuck at the "unclear specification" level by design, since they have no mechanism to receive explicit guidance at all.

---

> ### Author Response · Authors · 2025-11-20
> **Rebuttal for GTfZ (Part 3)**
>
> **Q2: How do you ensure RewardAnything learns generalizable principle-following rather than overfitting to LLM judge patterns?**
>
> This is an excellent methodological question. While our training data is synthetically generated using LLM judges, RABench itself was curated with extensive human effort to ensure it reflects genuine human judgment. We manually verified each instance in RABench, with two independent human annotators reviewing every element (principle, prompt, responses, and rankings). The inter-annotator agreement of 89% indicates good reliability, and we retained only instances where both annotators agreed on validity. This human-validated benchmark provides robust evidence of genuine principle-following capabilities beyond simply mimicking the training judge's patterns.
>
> Moreover, synthetic training data with human-validated evaluation is common and accepted practice in reward modeling research [2,3,4]. What ultimately matters is whether the model generalizes to human-validated test cases reflecting real human preferences, which our RABench results (Table 3) clearly demonstrate. RewardAnything achieves principle-following capabilities comparable to GPT-4.1, one of the most capable models available.
>
> Most critically, Section 6.3 provides end-to-end validation in a real downstream alignment task with real safety implications. The principle we used was manually crafted to be intentionally complex and outside our training distribution, as it contains multiple objectives and made specific requirements. We then used RewardAnything and Skywork as the sole reward source for GRPO RL alignment and evaluated on established safety benchmarks (XSTest) and quality benchmarks (MT-Bench). The significant improvements in both safety and quality metrics compared to using Skywork RM demonstrate that RewardAnything genuinely learns to follow novel human-specified principles and produces rewards that effectively guide LLM alignment in practice, not just on benchmarks.
>
> We sincerely hope these clarifications address your concerns regarding our comparison methodology, training complexity, and principle engineering. Given these clarifications and the additional evidence we've provided, we respectfully ask if you would consider increasing your scores. We would be grateful for any further feedback or questions you may have. Thank you very much for your thorough review and the opportunity to clarify these important aspects of our work.Retry
>
> [1] Failure Modes of Learning Reward Models for LLMs and other Sequence Models, ICML 2023 Workshop MFPL
>
> [2] Skywork-Reward: Bag of Tricks for Reward Modeling in LLMs, arXiv 2410.18451
>
> [3] RewardBench 2: Advancing Reward Model Evaluation, arXiv 2506.01937
>
> [4] RM-R1: Reward Modeling as Reasoning, arXiv 2505.02387

---

### Official Review · Reviewer_zuWy · 2025-10-31

**Soundness:** 2
**Presentation:** 2
**Contribution:** 2
**Rating:** 6
**Confidence:** 3

**Summary:**

The paper identifies two major flaws in current reward models: (1) limited generalization and adaptability to new tasks, and (2) the lack of explicit bias control and interpretability due to reliance on implicit preference learning. To address these limitations, the authors propose a principle-following reward model that extends beyond fixed reward criteria. Analogous to instruction-following large language models, this framework enables the reward model to interpret and generalize over arbitrary principles (P) — which may represent task specifications, ethical guidelines, or combinations of multiple criteria with defined priorities. The model is expected to handle unseen principles at varying levels of specificity and abstraction.

To systematically evaluate this capability, the authors introduce RABENCH, a comprehensive benchmark designed to test reward models on their ability to adapt evaluation criteria according to explicitly provided natural-language principles.

Building on this, they develop Reward-Anything, a reward model trained via on-policy reinforcement learning. The objective is to encourage the model to produce correctly ranked lists consistent with the guiding principles, thereby promoting both adaptability and principled generalization.

**Strengths:**

The paper presents a clear and coherent narrative — progressing logically from the motivation of improving reward model generalization to the development of RABENCH and the proposed principle-following reward model. The methodology is well-structured and easy to follow. The authors further strengthen their contribution by conducting on-policy training with GRPO, featuring an intuitive reward design. Experimental results demonstrate promising generalization of the learned reward model when applied to diverse downstream tasks.

**Weaknesses:**

Didn't report performance on other popular benchmark like RewardBench to show generalization, and some other popular work uses reward model for test time compute (verifier for best of N), reporting numbers on these would make the generalization statement more robust.

**Questions:**

1. Are there any existing reward models trained via reinforcement learning (RL) that the authors compare their approach against? It would be helpful to understand how the proposed method performs relative to prior RL-based reward models in terms of generalization, adaptability, and bias control.

2. Could you report the REWARDANYTHING on REWARD bench?

3. How does instruction following capability correlates with the principle adherence capability, if we start with a IF tuned model (verifiy-if) would that results in even better performance?

4. How could we combine this with rubrics based rewards ideas? The model seemed to follow a set of principles which are analogous to rubrics (checks) which gains popularity recently for evaluating the response quality.

---

> ### Author Response · Authors · 2025-11-20
> **Rebuttal for Reviewer zuWy (Part 2)**
>
> **Q1: Comparing with RL-trained reward models:** Great question! We actually do compare against RM-R1 in Table 2 of our paper, which is also trained with GRPO and represents concurrent work in this space. We compare multiple variants and sizes of RM-R1 and consistently outperform even their largest 32B version. RM-R1 uses GRPO with reasoning SFT to enable reasoning before making pairwise judgments, making it a strong and relevant baseline for our approach. However, they do not involve any explicit principles as guidance, thus still suffering from the bias issues we aim to address.
>
>
>
> **Q2: Regarding instruction-following and principle adherence correlation:** This is a fascinating research question. While we didn't explore starting from instruction-tuned models like Verify-IF in this work, your intuition seems sound. Instruction-following capability could provide a strong foundation for principle adherence. We believe this would be an excellent direction for future work and appreciate the suggestion.
>
>
>
> **Q3: Regarding rubrics-based rewards:** You've identified an important connection! Our approach is indeed fundamentally related to rubrics-based evaluation. We focus on two key contributions in this space: (1) systematically evaluating general purpose LLMs and reward models' ability to generalize across different rubrics or principles through our RABench benchmark, and (2) developing a reward model trained via reinforcement learning that can follow diverse principles effectively. While there has been prior work on rubrics-based evaluation, we believe our emphasis on generalization to unseen principles and the use of RL for training principle-following capabilities represents a novel contribution. If you feel there's any specific work in this area that we might have missed or should discuss more prominently, we would be happy to clarify the connections and distinctions.
>
> We hope these responses adequately address your concerns and demonstrate the robustness of our approach across diverse evaluation scenarios. The RewardBench-v2 results in particular provide strong evidence for the generalization capabilities you highlighted as important. We would greatly appreciate knowing whether these clarifications and additional results have addressed your concerns, and whether they might warrant increasing the score. Your feedback has been invaluable in strengthening our work, and we remain open to any further suggestions you might have.

---

> ### Author Response · Authors · 2025-11-20
> **Rebuttal for Reviewer zuWy (Part 1)**
>
> We sincerely thank you for the positive feedback and constructive suggestions. We're delighted that you found our work well-structured with promising generalization results.
>
> **W1 / Q2: Regarding RewardBench performance and generalization:** Thank you for this excellent suggestion. We initially did not include RewardBench at the time of writing because the original benchmark was experiencing severe data contamination issues as indicated by RewardBench authors (https://huggingface.co/datasets/natolambert/skyworks-rewardbench-contamination), with multiple smaller models achieving over 90% accuracy after confirmed training on test data. However, RewardBench-v2 was released afterwards as a significantly harder, cleaner benchmark. We are now reporting the results on RewardBench-V2, on models released before the benchmark, thus there should be no concerns on data contamination for these results.
>
> | **Model**                        | **Type**       | **Overall** | **Factuality** | **Precise IF** | **Math** | **Safety** | **Focus** | **Ties** |
> | -------------------------------- | -------------- | ----------- | -------------- | -------------- | -------- | ---------- | --------- | -------- |
> | Claude Sonnet 3.7 (20250219)     | Generative     | 75.4        | 73.3           | 54.4           | 75.0     | 90.3       | 92.1      | 67.2     |
> | **RewardAnything-8B**            | Generative     | 74.5        | 66.7           | 47.5           | 80.3     | 84.9       | 84.2      | 83.3     |
> | GPT-4.1 (20250414)               | Generative     | 72.3        | 82.9           | 39.7           | 65.2     | 87.3       | 73.4      | 85.4     |
> | Skywork-Reward-Llama-3.1-8B-v0.2 | Discriminative | 71.8        | 69.7           | 40.6           | 60.1     | 94.2       | 94.1      | 71.7     |
> | GPT-4o (20240806)                | Generative     | 64.9        | 56.8           | 33.1           | 62.3     | 86.2       | 72.9      | 78.2     |
> | GPT-4o-mini (20240718)           | Generative     | 58.0        | 41.1           | 34.4           | 51.9     | 76.7       | 74.1      | 69.6     |
> | GRM-llama3-8B-sftreg             | Discriminative | 60.9        | 61.9           | 38.8           | 57.9     | 78.7       | 68.3      | 59.8     |
> | ArmoRM-Llama3-8B-v0.1            | Discriminative | 66.5        | 65.7           | 41.9           | 66.1     | 82.2       | 76.6      | 66.3     |
>
> Here, we compare the best generative models and discriminative reward models under special tuning, that was all released before the RewardBench-v2 dataset, so the table does not involve any possibility for us and for all candidates here to be contaminated. Our model performs between Claude Sonnet 3.7 and GPT-4.1, which are considered one of the best general purpose, close LLMs that’s publically available.
>
> Examining the breakdown reveals interesting patterns. While we fall somewhat short on factuality compared to GPT-4.1, we achieve substantially better performance on precise instruction following, mathematics, and focus. This suggests our principle-following approach excels particularly in domains requiring careful adherence to specified criteria, which aligns perfectly with our design philosophy. For factuality evaluation, this suggests hybrid rewards that combines sources of facts like RAG-enhanced reward might be necessary and could be a great direction for future work.

---

> > ### Comment · Reviewer_zuWy · 2025-11-26
> >
> > Thanks for your comprehensive responses, it addresses my concerns. And I'll maintain my recommendation / score to accept the paper.

---

### Meta-Review · Area_Chair_hmdS · 2025-12-16

**Summary:**

This paper proposes RewardAnything, a reward model designed to follow dynamically provided natural language principles rather than being locked into fixed preference distributions learned from static datasets. The method uses Group Relative Policy Optimization (GRPO) to train models that can adapt to diverse evaluation criteria at inference time. The paper also introduces RABench, a benchmark for evaluating principle-following generalization. Reviews were mixed with initial ratings of 2, 4, and 6, with one reviewer (rLRA) eventually raising their score from 2 to 6 after extensive discussions, resulting in final scores of 6, 4, and 6, as claimed by the authors (Reviewer rLRA indeed confirmed to raise the score, but I cannot tell whether it was to 4 or 6).

The primary strengths include addressing an important limitation of current reward models—their rigidity and inability to adapt to new criteria without retraining. The method demonstrates state-of-the-art performance on traditional RM benchmarks when given well-defined principles.

However, significant concerns remain about novelty and methodological clarity. Reviewers noted the similarity to prior work on context-aware preference modeling (NeurIPS 2024) and deliberative alignment (OpenAI 2024), though none released model weights for direct comparison. I tend to agree with reviewers that the work also misses important comparisons that combine the idea of context-aware reward models with mixture modeling, such as [1].

[1].    MiCRo: Mixture Modeling and Context-aware Routing for Personalized Preference Learning, EMNLP 2025
[2].    Test-time alignment via hypothesis reweighting, https://arxiv.org/abs/2412.08812

The core architectural design (principle-conditioned generative RM with listwise data and RL objective) appears incremental over existing instructable reward modeling approaches, with the main distinction being GRPO training methodology. Given the above concerns, I recommend reject, but encourage the authors to address the novelty and comparison issues in a future revision.

**Reviewer Concerns:**

Two of them have been addressed, one outstanding

**Reviewer Scores:**

One of the reviewers agreed to improve the score from 2, but it's unclear what the new score will be.

---

### Decision · Program_Chairs · 2026-01-26

Reject